# Multisystem Inflammatory Syndrome in Children (MIS-C), Possibly Due to COVID-19 mRNA Vaccination

**DOI:** 10.3390/vaccines11050956

**Published:** 2023-05-06

**Authors:** Alije Keka-Sylaj, Atifete Ramosaj, Arbana Baloku, Leonora Zogaj, Petrit Gjaka

**Affiliations:** 1Institute of Anatomy, Faculty of Medicine, University of Prishtina, 10000 Prishtina, Kosovo; atifete.ramosaj@uni-pr.edu; 2Pediatric Clinic, University Clinical Center of Kosovo, 10000 Prishtina, Kosovo; arbana.baloku@gmail.com (A.B.); zogaj.nora@gmail.com (L.Z.); petritgjaka@gmail.com (P.G.)

**Keywords:** COVID-19 vaccination, multisystem inflammatory syndrome in children (MIS-C), multisystem inflammatory syndrome following SARS-CoV-2 vaccination (MIS-V), SARS-CoV-2 IgG spike antibodies, children

## Abstract

Multisystem inflammatory syndrome in children (MIS-C) is a potentially life-threatening childhood disease caused by SARS-CoV-2 infection, manifested by the persistence of fever and multi-organ dysfunction, elevated inflammatory markers, and the lack of an alternative diagnosis. It is still unknown if vaccination can precipitate or abrogate MIS-C or if a natural infection preceding or occurring at the time of vaccination plays any role. We present one case of MIS-C in a 16-year-old girl who was fully immunized against COVID-19 (Pfizer), with the second dose received three weeks prior to onset of the disease. She had no history of COVID-19 disease or contact with COVID-19 patients. At admission, she was somnolent, pale, and dehydrated, with cyanotic lips and cold extremities; she was hypotensive with tachycardia and poorly palpable pulses. Initial laboratory results revealed elevated levels of inflammatory markers, and high level of SARS-CoV-2 IgG spike antibodies, while testing for SARS-CoV-2 acute infection and other inflammatory etiologies were negative. Vaccine-related MIS-C was suspected in our case due to the development of MIS-C three weeks following the second dose of the COVID-19 mRNA vaccine, the absence of previous infection or exposure to SARS-CoV-2, and a positive result for IgG anti-spike (S) antibodies.

## 1. Introduction

The multisystem inflammatory syndrome in children (MIS-C) is a severe postinfectious hyperinflammatory condition that generally manifests 2–6 weeks after a typically mild or asymptomatic infection with SARS-CoV-2, with fever and multisystem organ involvement [1]. Since 2020, a large number of MIS-C cases have been recorded globally with a variety of clinical presentations, the majority of which include a persistent fever, elevated inflammatory markers, and the emergence of multi-organ failure, in the absence of an alternative diagnosis [2,3,4].

Fortunately, vaccination offers excellent protection against severe COVID-19, but some researchers worry that immunization against SARS-CoV-2 may trigger MIS-C/A.

Due to the fact that mRNA vaccines are now authorized for use in children over the age of six months, certain cases of MIS-C have lately been documented after mRNA immunization following SARS-CoV-2 vaccination (MIS-V) [5,6,7]. It is considered a rare entity although its exact incidence, prevalence, and pathophysiology are still unknown [8]. Furthermore, it is still unclear if the vaccination can precipitate or abrogate MIS-V or if a natural infection preceding or occurring at the time of vaccination plays any role [9]. It has been suggested that a prior asymptomatic or symptomatic COVID-19 infection could induce the dysregulation of T cell responses, cytokine storm, and/or immune hyper-reactivity [8].

We present the case of a 16-year-old Albanian girl who manifested MIS-C three weeks after receiving the second dose of the COVID-19 vaccine (Pfizer BioNTech).

### Case Report 

In November 2021, a critically ill 16-year-old girl was admitted to the pediatric intensive care unit at the University Clinical Center of Kosovo.

She presented fatigue, fever, body pain, a loss of appetite, nausea, vomiting, and diarrhea two days before admission, and was treated in the primary health care clinic with intravenous fluids and antipyretics. On admission, she was somnolent, pale, and dehydrated, with cyanotic lips and cold extremities. Her blood pressure (79/38 mmHg) was only measured after a rapid fluid infusion. Her heart rate (HR) was over 100 beats/min with poorly palpable pulses, and her respiratory rate (RR) was greater than 45 breaths/min, with an oxygen blood saturation (SpO_2_) of 93% and a nasal O_2_ flow of 3–5 L/min.

Her lungs were clear; she had no murmur or cardiac friction rubs. The abdomen was soft and not painful with palpation, and no hepatosplenomegaly was noted. The skin was too pale, with cold extremities and no rash or edema. The lips were dry, the tongue was layered, and there was oropharyngeal hyperemia without cervical and submandibular lymphadenopathy. She had previously been healthy; her body weight was 55 kg and her height was 167 cm; she had no prior history of COVID-19 disease or positive contact with COVID-19-infected individuals or acquaintances who were symptomatic. There was no personal or family history of allergic reactions, vasculitis, autoimmune disorders, cardiac disease, diabetes, or hereditary disease, and this was her first hospitalization. 

During this period, according to National Institute for Public Health in Kosovo, the number of positive COVID-19 cases in Kosovo had greatly dropped, with only 10 to 15 positive cases per day from hundreds that were tested. A large proportion of the population were immunized (850,415 with the first dose and 757,103 with the second dose) and vaccination had also begun in children over the age of 16. 

Our patient had received two doses of the COVID-19 vaccine (Pfizer Germany), the first dose three months and the second dose three weeks prior to the onset of symptoms, respectively. Therefore, as the nasopharyngeal SARS-CoV-2 PCR test result was negative too, we ordered specific measurements for SARS-CoV-2 antibodies. The serology revealed low levels of anti-SARS-CoV-2 nucleocapsid IgG antibody (2.6), but high levels of anti-SARS-CoV-2 spike IgG (>2500). 

Initial laboratory results revealed high levels of inflammatory markers, including 234 mg/L of C-reactive protein (CRP), an erythrocyte sedimentation rate (ESR) of 70 mm/h, 5.66 ng/mL of procalcitonin (PCT), and 56.68 pg/mL of Interleucin 6, and there was leukopenia with lymphocytosis. In addition to hypoalbuminemia and hypoproteinemia, there was an elevation of alanine aminotransferase (ALT), aspartate aminotransferase (AST), lactic dehydrogenase (LDH), and creatine kinase (CK). Within 24 h, d-dimer increased from 222 ng/mL to 509 ng/mL (ref. range 200), while all other coagulation tests were within normal limits. The capillary blood electrolytes (Na, K, and Ca) and gases (pO_2_ and pCO_2_) were nearly normal.

The urinalysis revealed the presence of proteinuria (30–100 mg/dL), nonsignificant leukocyturia, erythrocyturia, and bacteriuria, with two urine cultures being negative, and no other obvious microbial cause of inflammation was found, including a nasopharyngeal culture and two blood cultures for bacterial sepsis. We additionally performed tests for toxoplasmosis, rubella, cytomegalovirus, and herpes simplex, which were all negative, but we did not exclude other viral infections, such as influenza, adenoviruses, or enteroviruses. Therefore, we cannot rule out the possibility that an undetected infection contributed to this case of MIS-C following mRNA vaccination. Clinical and laboratory findings are shown in Table 1. 

The Ph acid base of the blood, pCO_2_ (the partial pressure of carbon dioxide), and pO_2_ (the partial pressure of oxygen) was balanced (Na—sodium, K—potassium, Ca—calcium, HCO_3_—bicarbonate, and BE—base excess).

Electrocardiography (ECG), echocardiography, and chest X-rays were all performed. The first ECG shows a slight elevation in the ST segment in leads I and II (Figure 1), whereas echocardiography shows ventricular dysfunction with a shortening fraction (FS) of 24% and an ejection fraction (EF) of 44%.

The chest radiography showed the silhouette of the heart that was slightly enlarged, as well as the congestion of pulmonary vessels, as shown in Figure 2.

Her clinical condition worsened about 8 h after admission; she had chest pains and hypotension, so a second ECG, echocardiography, and cardiac enzymes were performed. The second ECG revealed pericardial lead ST segment elevation (Figure 3), whereas echocardiography revealed myocardial hypocontractility with a severe decrease in ventricular ejection fraction below 30%. The cardiac enzymes were elevated compared to the first measurement, as follows: creatine kinase—from 32 to 201 (ref. range: 30–135 units/L), creatine kinase MB—from 10 to 44 (ref. range: 0–7 mcg/L), and troponin I—from 0.18 to 6.37 (ref. range: 0–0.04 ng/mL). Therefore, a continuous infusion of noradrenalin was initiated at 4 mcg/kg/min (with reductions up to 2 mcg), which improved her blood pressure and heart rate.

After about 16 h, the child’s condition worsened, with decreased blood saturation and signs of heart failure with pulmonary vascular congestion, as seen on the chest X-ray (Figure 4). As a result, supportive respiratory care was provided for 24 h using continuous positive airway pressure (CPAP), and rehydration was concurrently maintained using fluids, diuretics, and inotropes (dopamine and noradrenalin).

Since clinical, laboratory, and additional examinations revealed that the patient met the MIS-C criteria, we initiated treatment with corticosteroids (2 mg/kg) and immunoglobulin (1 g/kg). The anticoagulation therapy with enoxaparin was also administered due to high D-dimer levels. About 48 h later, inflammatory parameters began to decrease and the condition gradually improved; however, inotropic support with continuous reduction was required for ten days due to low ventricular ejection fraction and hypotension. The pulmonary function was improved, and blood oxygen saturation was effectively maintained with nasal O_2_. The third chest X-ray was slightly better than the second one, as shown in Figure 5.

On the 16th day of the hospital stay, almost all laboratory tests were within the normal range, as well as the electrocardiogram (Figure 6), echocardiography, and lung and abdominal ultrasound examinations, so the child was discharged home with the recommendation of using aspirin 80 mg/day for four weeks.

During follow-up outpatient visits, the child’s general condition remained stable, and the blood tests, including cardiac enzymes, as well as ECG and echocardiography were normalized. The SARS-CoV-2 IgG antibodies were reduced to 2.1 (ref. range >1.0) after a week, whereas SARS-CoV-2 IgG spike antibodies remained high at >2500, as shown in Table 1.

## 2. Discussion

Since April 2020, when the first cases of an unusual inflammatory illness in children were reported in the United Kingdom [10] and Italy [11], there have been additional reports from many countries around the world. Reported MIS-C cases manifested a wide spectrum of signs and symptoms due to multisystem organ dysfunction, including cardiac, renal, respiratory, hematologic, gastrointestinal, dermatologic, and neurological dysfunction. Due to severity of the disease with myocardial injury, toxic or hypotensive shock, and multiorgan failure, some of them required intensive care, and, unfortunately, a few of them died [12].

Since the vaccines against COVID-19 were approved for adults and children, millions of them have been immunized. Fortunately, vaccination against COVID-19 is a powerful tool that has decreased morbidity and mortality rates around the world, and the most frequently reported adverse events associated with vaccination were non-serious potential allergic reactions [13].

The rates of all forms of adverse events were considerably higher after receiving the Pfizer-BioNTech vaccine, and, in addition to mild side effects, serious adverse events were also reported, including anaphylaxis, thromboembolic events, and acute hypertension [14].

Moreover, myocarditis and pericarditis were the most common serious adverse effects of the Pfizer-BioNTech vaccine, primarily in children over the age of 12 [15]. 

Our case initially had fatigue, fever, body pain, a loss of appetite, nausea, vomiting, and diarrhea progressing to critical conditions, with signs and symptoms of toxic shock syndrome. She developed chest pain and decreased ventricular function, as confirmed by ST segment elevation on the electrocardiogram (EKG); elevated cardiac enzymes; left ventricular systolic dysfunction, as confirmed by the echocardiography; and the need for inotropic drug support. The patient met the criteria for MIS-C based on clinical manifestations, elevated inflammatory markers, and negative multiple investigations for infectious and inflammatory etiologies. It was presented in the context of a significant decrease in positive COVID-19 cases in Kosovo for four months in a row. Therefore, due to the onset of the disease, three weeks following the vaccination, a negative nasopharyngeal SARS-CoV-2 PCR test, and IgM SARS-CoV-2 antibodies, we suspected a vaccine-induced anti-body response rather than a SARS-CoV-2 infection-induced antibody response.

This is supported by high levels of SARS-CoV-2 spike protein IgG antibodies, consistent with vaccine immunization versus slightly positive IgG anti-nucleocapsid antibodies, possibly due to an early prior infection, even though no known exposure was available to inform the timing of the infection. Although anti-nucleocapsid antibodies are indicative of past or recent SARS-CoV-2 infection, anti-spike protein antibodies can be induced either by SARS-CoV-2 infection or COVID-19 vaccination [16,17]. In our case, the vaccination was only three weeks before, and, moreover, the level of anti-nucleocapsid antibodies was slightly positive, but dropped after a week.

As MIS-C is associated with SARS-CoV-2 infection, the CDC and the FDA included MIS-C on a list of adverse events of special interest for COVID-19 vaccine safety monitoring after the emergency use authorization of COVID-19 vaccines [16,18]. 

The risk of hyperinflammatory syndrome following the COVID-19 mRNA vaccine was significantly lower compared with the post-SARS-CoV-2 MIS-C reported in the study of a large group of children aged 12 to 17 in France who received the vaccine [19].

From 2028 reported cases with adverse reactions relating to the COVID-19 mRNA vaccine, 102 of them had myocarditis or pericarditis, while 12 children manifested a hyperinflammatory syndrome [19]. 

MIS-C following SARS-CoV-2 mRNA vaccination was reported in 21 individuals aged 21 and younger, 6 of whom had no prior evidence of COVID-19 infection [6].

Similar to our case, several reports in the previous two years suggested a possible link between the mRNA COVID-19 vaccine and the systemic hyperinflammatory syndrome seen in COVID-19 patients [9,20,21,22,23].

However, the incidence of MIS-C after natural SARS-CoV-2 infection is much higher (estimated at approximately 200 per million children) than the incidence of MIS-C after COVID-19 vaccination, which is notably low with 1.0 per million cases [24]. Therefore, the benefits of vaccination outweigh the known risks in this regard [24]. 

## 3. Conclusions

Despite the fact that the findings suggest that MIS-C without evidence of SARS-CoV-2 infection is uncommon, vaccine-related MIS-C was suspected in our case due to the development of MIS-C three weeks after the second dose of COVID-19 mRNA vaccine, the absence of previous SARS-CoV-2 infection or exposure, and a very high level of anti-spike IgG antibodies. This is also based on the very low incidence of positive cases in our country for several months during that time period.

Our findings raise the possibility that vaccination caused MIS-V, but it is also possible that another viral infection, such as influenza, adenoviruses, or enteroviruses, could have triggered this condition. Moreover, it should be emphasized that MIS-C following mRNA vaccination is extremely rare.

## Figures and Tables

**Figure 1 vaccines-11-00956-f001:**
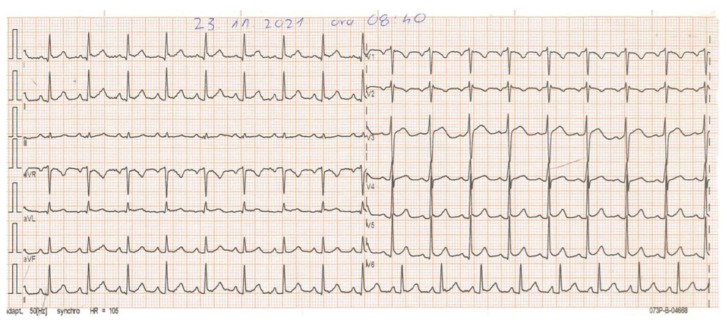
ECG on admission. The heart rate of 106 beats per minute, a left axis, sinus rhythm, and a slight elevation in the ST segment in leads I and II.

**Figure 2 vaccines-11-00956-f002:**
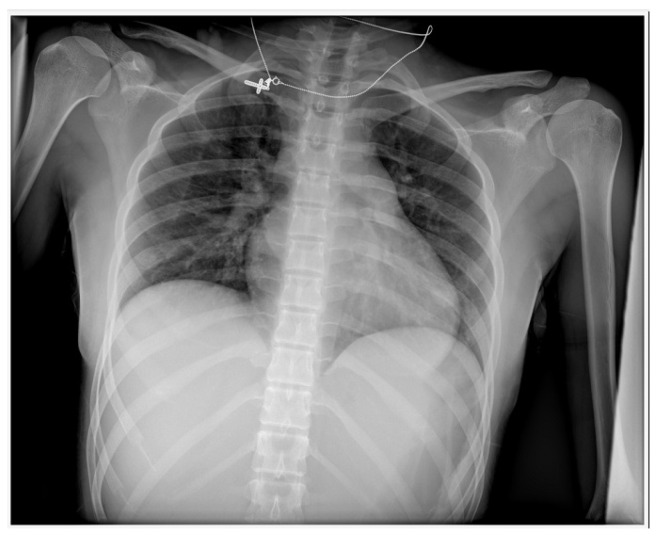
Chest X-ray in AP projections: slightly enlarged silhouette of the heart; pulmonary vessel congestion; no signs of lung consolidation or atelectasis; no pleural effusion or pneumothorax either.

**Figure 3 vaccines-11-00956-f003:**
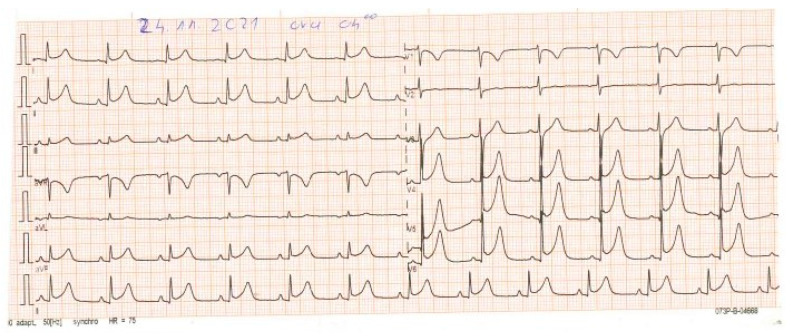
The second ECG 8 h later. HR of 75 beats/minute, left axis, sinus rhythm, and ST segment elevation in the precordial leads.

**Figure 4 vaccines-11-00956-f004:**
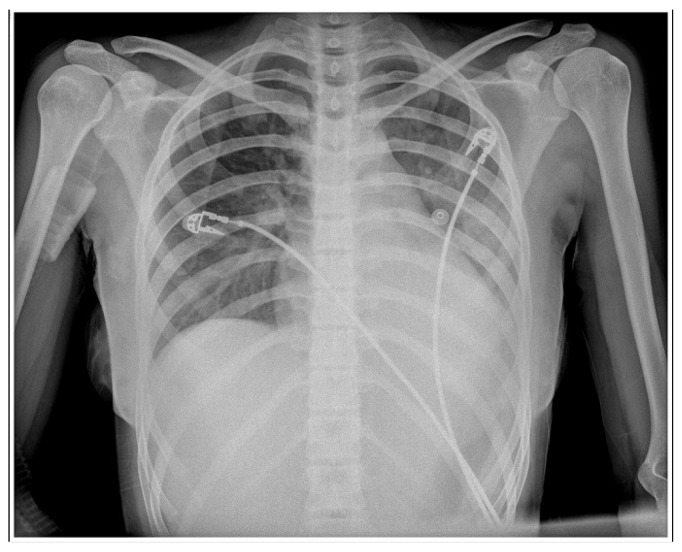
Second frontal chest radiography shows the progression of pulmonary vascular congestion as well as atelectasis of the left lower lobe as a result of a cut-off of the lower left lobar bronchus (mucus plug).

**Figure 5 vaccines-11-00956-f005:**
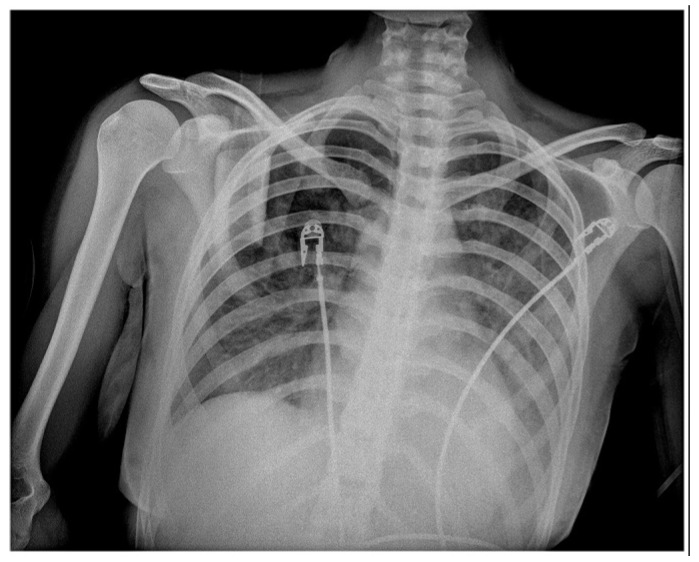
The third frontal chest radiography shows that the pulmonary vascular congestion is less evident.

**Figure 6 vaccines-11-00956-f006:**
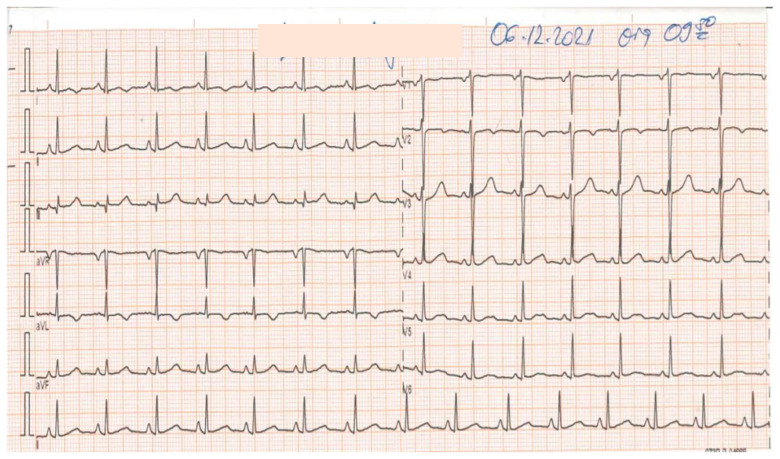
The ECG records a normal sinus rhythm, ST segment, QRS complex, QT segment, and Q wave with a normal amplitude and interval.

**Table 1 vaccines-11-00956-t001:** The table shows the demographic and clinical characteristics of the patient on admission, as well as laboratory findings on admission and during the hospital stay.

Variable	Patient					
Age (years)	16					
Sex	F					
Body weight	55 kg					
Height	167 cm					
Initial manifestations	fatigue, fever, body pain, a loss of appetite, nausea, vomiting, and diarrhea					
Manifestations at admission	somnolent, pale, and dehydrated, with cyanotic lips and cold extremities,					
Blood pressure	79/38 mmHg					
Heart rate	100 beats/min					
Respiratory rate	45 breaths/min					
Oxygen blood saturation (SpO_2_)	93% with O_2_ flow of 3–5 L/min					
Laboratory findings	Day 1		Day 2	Day 3	Day 8	Day 14
ESR (5–10)	70			42		16
CRP (0.0–6.0 mg/L)	234	186	143	70.9	20	6.4
PCT (0.0–0.5 ng/mL)	5.66	6.73		1.32		0.73
Interleukin 6 (<7.0 pg/mL)		56.68				
Ferritin (15–150 ng/mL)		269.1				
Cholesterol (3.60–5.70 mmol/L)		3.2			4.76	
Triglicerides (0.45–1.81 mmol/L)		0.75			2.46	
Urea (1.70–8.30 mmol/L)	6.23	6.43	7.27	8.78		9.88
Uric Ac (155–430 umol/L)	327					
Creatinine (53–115 umol/L)	109	115	96.1	105.1		90.6
Albumin (35.0–52.0 g/L)	33.6	39.0	33.8	33.6	32.9	
Total protein (64.0–83.0 g/L)	60.5	62.2	70.3	78.1	68.6	67
Glycaemia (3.6–6.4 mmol/L)	8.05	11.69	11.69	12.54		6.61
Total bilirubine (3.6–6.4 mmol/L)	21.4	10.2				
Direct bilirubine (0–5.1 umol/L)	4.8	3.49				
ALT (3–41 U/L)	165	213	35	39	20	29
AST (2–37 U/L)	124	143	74	80	25	45
ALP (43–115 U/L)	29		29	31	53	56
GGT (3–55 U/L)		16				
CK (38–171 U/L)	32	201/430	180	46	17	42
CK-MB (5–25 U/L)	10	44				
Troponina I (0–0.04 ng/m)	0.18	6.37				
LDH (230–460 U/L)	202	396	671	732	521	
Complete blood cell count (CBC)						
WBC (3.5–10)	2.7	6.5	15.1	14.6	11.6	7.4
RBC (3.8–5.8)	4.70	6.07	5.64	5.95	5.06	4.58
HGB (11.0–16.5)	12.2	16.2	14.8	134	13.3	11.3
HCT (35.0–50.0)	44.1	57.7	53.4	48.3	47.8	42.4
PLT (150–390)	122	125	145	132	131	222
LYM (17.0–48.0%)	42.3	22.3	7.5	15.3	30.4	26.1
GRA (43.0–76.0%)	45.8	67.0	89.5	81.9	65.3	69.4
Capillary blood gases and electrolytes						
Ph (7.35–7.45)	7.47	7.46	7.49	7.53	7.45	7.4
pCO_2_ (34–46 mm Hg)	27	33	35	34	39	32
pO_2_ (80–105 mm Hg)	51	51	57	68	54	74
Na^+^ (130–145 mmol/L)	129	130	134	139	137	135
K^+^ (3.4–5.1 mmol/L)	4.5	3.8	3.5	3.5	2.6	3.5
Ca^++^ (1.17–1.24 mmol/L)	1.15	1.21	1.17	1.15	1.19	1.19
HCO_3_ (22–26 mmol/L)	19.7	23.5	26.7	28.4	27.1	25.3
BE (−4 to +2 mmol/L)	−4	−0.3	3.4	5.7	3.1	2.2
Coagulation tests						
D-dimer (˂200)	222	599	155	117	55	44
PT (70–120%)			73%			
INR (099–1.2)			1.2			
PTT (25–35″)			25″			
TT (12–22)			18″			
Protrombina			100%			
Proakcelerina			106%			
Prokonvertina			72%			
F VIII			105%			
F IX			103%			
F X			88%			
F XI			90%			
F XII			70%			
The serology for SARS-CoV-2 and other infections						
					Day 20	Day 20
IgM SARS-CoV 2 (>1.1 positive)	0.4					2.1
IgG SARS-CoV 2 (>1.1 positive)	2.6				2.1	>2500
Anti SARS-CoV2 spike (>1.1)	>2500				>2500	
IgM rubeola			Negative			
IgG rubeola			Positive			
IgM HSV			Negative			
IgG HSV			Positive			
IgM toxoplasma			Negative			
IgG toxoplasma			Positive			
IgM CMG			Negative			
IgG CMV			Positive			
Bacterial culture tests						
Throat culture	Sterile; twice					
Blood culture	Sterile; twice					
Urine culture	Sterile; twice					

CRP: C-reactive protein; ESR: erythrocyte sedimentation rate; PCT: procalcitonin; ALT: alanine aminotransferase; AST: aspartate aminotransferase; ALP: alkaline phosphatase; GGT: gamma-glutamyl transpeptidase; CK: creatine kinase; CK-MB: creatine kinase MB; LDH: lactic dehydrogenase WBCs: white blood cells, RBCs: red blood cells, HGB: hemoglobin, HCT: hematocrit, PLTs: platelets, LYMs: lymphocytes, GRAs: granulocytes.

## Data Availability

Not applicable.

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
