# Peer review of "Multisystem Inflammatory Syndrome in Children (MIS-C), Possibly Due to COVID-19 mRNA Vaccination"

_vaccines, 2023, doi:10.3390/vaccines11050956_

Round 1

Reviewer 1 Report

The authors reported a 16-year-old girl suffered from MIS-C, while she had no history of COVID-19 infection and received the send dose of COVID-19 mRNA vaccine three weeks ago. The authors’ efforts to describe all details about the patient’s symptom and related treatment should be appreciated. However, there is basically no evidence to link the MIS-C with mRNA vaccine. The only evidence is that the MIS-C was developed three weeks after mRNA vaccination. However, I personally think, these symptoms could also be induced by infection of any viruses. Further, as authors cited in the manuscript, many similar cases in children has been reported. Combined with the facts that only one case was studies, and that no sufficient evidence to link the MIS-C syndrome and COVID-19 mRNA vaccination was provided, this manuscript, in my own opinion, lacks novelty.

Author Response

Response: Dear Editor, Thank you for reading our case study and providing feedback and recommendations.

In this case study, we linked the MIS-C with the mRNA vaccine in this case based on:

She had no personal or family history of infectious diseases, including COVID-19 or other viral infections.

She had no history of COVID-19 disease, contact with a COVID-19 patient, or symptomatic acquaintances.

During this period, according to the National Institute for Public Health in Kosovo, the number of positive COVID-19 cases in Kosovo had greatly dropped.

She had been fully vaccinated with the COVID-19 vaccine (Pfizer), with the first dose received three months prior and the second three weeks prior to the onset of symptoms.

She initially had fatigue, fever, body pain, loss of appetite, nausea, vomiting, and diarrhea, progressing to a critical condition with signs and symptoms of toxic shock syndrome; therefore, she required hospitalization.

She had evidence of systemic inflammation with elevated inflammatory markers such as the erythrocyte sedimentation rate (ESR), C-reactive protein (CRP), procalcitonin (PCT), and interleukin 6.

She had cardiac involvement indicated by the ECG, a left ventricular ejection fraction <55%, and cardiac enzymes above the laboratory normal range.

The nasopharyngeal SARS-CoV-2 PCR test resulted negative.

We measured SARS-CoV-2 antibody levels, which revealed a high level of anti-SARS-CoV-2 spike IgG (>2500), which remained high (their assay was used to monitor the vaccine-induced humoral response), and a low level of anti-SARS-CoV-2 nucleocapsid IgG antibody (2.6), which decreased to 2.1 (ref. range >1.0) after a week.

The CDC's case definition for MIS-C included also a positive serology test regardless of COVID-19 vaccination status.

"Detection of anti-nucleocapsid antibody or anti-spike protein antibody fulfill criteria for the case definition, when feasible SARS-CoV-2 anti-nucleocapsid antibody testing is recommended, particularly in children with a history of COVID-19 vaccination because anti-nucleocapsid antibody is indicative of SARS-CoV-2 infection, while anti-spike protein antibody may be induced either by COVID-19 vaccination or by SARS-CoV-2 infection".

We also performed tests for other infectious and inflammatory aetiologies (throat culture, blood culture, urine culture), as well as serology for toxoplasmosis, rubella cytomegalovirus, and herpes simplex, which were negative.

We appreciate and agree with your opinion that these symptoms could also be induced by other viruses, but we didn’t have any other cases with viral infections like influenza, adenoviruses, or enteroviruses during that time period, so we didn’t test for those viral infections.

During the COVID-19 pandemic, we had other MIS-C cases after natural infection with SARS-CoV-2 infection, but we didn't have other cases of MIS-C after vaccination; therefore, this is only a case study.

Fortunately, vaccination against COVID-19 was a powerful tool that has decreased morbidity and mortality worldwide, and the incidence of MIS-C after natural SARS-CoV-2 infection was much higher than the incidence of MIS-C after COVID-19 vaccination.

We believe that this case report is appropriate for publication in the journal Vaccines because this condition is still a rare entity, and still unknown are the exact incidence, prevalence, and pathogenesis; therefore, it will add important additional information about this condition.

We appreciate your opinion, hope that we made proper corrections, and remain hopeful that this modest case report will be considered for publication in your precious journal.

Best regards, Alije Keka

Reviewer 2 Report

The effect of vaccination on MISC-C is discussed in the presented case report.

Minor changes

·       MIS-C abbreviation should be revised in the article and all MISC should be corrected as MIS-C.

·       In line 146, the authors should explain why noradrenaline was initiated at a dose of 4 mcg/kg/min.

·       In Figure 6, the patient's name is written on the ECG. For ethical reasons, this name should be removed.

Author Response

Response: Dear Editor, Thank you for reading my paper and providing feedback and recommendations.

  • All MISC abbreviations were replaced and corrected with MIS-C abbreviation
  • Our patient was 16 years old and weighed 55 kilograms. She had hypotension on admission, which improved slightly after the rapid giving of intravenous fluid administration but dropped again about 8 hours later when her clinical condition worsened with chest pain, bradycardia, and hypotension.

We administered noradrenaline according to Medscape dose recommendations for adults.

Initial: 8–12 mcg/min IV infusion; titrate to effect

Maintenance: 2-4 mcg/min IV infusion

  • I apologize for this technical issue, I inserted the box above the name, but it was shifted when I attached Figure 6 of the ECG. I corrected it.

We appreciate your opinion, hope that we made proper corrections, and remain hopeful that this modest case report will be considered for publication in your precious journal.

Best regards, Alije Keka

Reviewer 3 Report

The case report’s data were good and interesting. The way of representationnis also good. No much comments, except references. Even though MISC and COVID are new, but in this 3 yrs there are so many research and review articles already published. So author should find more relevant references to site in this case report to improve the impact of this publication. 

Author Response

Response: Dear Editor, Thank you for reading our paper and providing feedback and recommendations.

We appreciate your reasonable recommendations, and therefore we cited and added additional relevant references.

We appreciate your opinion, hope that we made proper corrections, and remain hopeful that this modest case report will be considered for publication in your precious journal.

Best regards, Alije Keka

Round 2

Reviewer 1 Report

To avoid offering misleading information to the society, the authors should rephrase the manuscript and emphasize that, even this MISC case was due to mRNA vaccination, which was actually not confirmed in this study, the MISC after mRNA vaccination is extremely rare.

Author Response

Dear Editor,

Thank you for reading our case study and providing feedback and recommendations.

We appreciate your suggestions, with which we completely agree.

Therefore, during the second revision, the following changes have been made to the introduction section, case report:

In paragraphs 89-92, was added; We additionally performed tests for toxoplasmosis, rubella, cytomegalovirus, and herpes simplex, which were all negative, but we didn't exclude other viral infections like influenza, adenoviruses, or enteroviruses because no cases of such infections were reported in our hospital during that period.

In the conclusion section, the following changes have been made:

Paragraphs 249-251 were deleted.

In paragraphs 253-257, was added; Our findings raise the possibility that vaccination caused MIS-V, but it is also possible that another viral infection such as influenza, adenoviruses, or enteroviruses could have triggered this condition, which was not confirmed in our study. We consider that MIS-C following mRNA vaccination is extremely rare; therefore, all possible causes of this condition should be excluded.

We hope that we made proper corrections, and remain hopeful that this modest case report will be considered for publication in your precious journal.

Best regards, Alije Keka